# A Compact Memristor Model Based on Physics-Informed Neural Networks

**DOI:** 10.3390/mi15020253

**Published:** 2024-02-08

**Authors:** Younghyun Lee, Kyeongmin Kim, Jonghwan Lee

**Affiliations:** Department of System Semiconductor Engineering, Sangmyung University, Cheonan 31066, Republic of Korea; 2022d2004@sangmyung.kr (Y.L.);

**Keywords:** memristor, physics-informed neural network (PINN), Verilog-A

## Abstract

Memristor devices have diverse physical models depending on their structure. In addition, the physical properties of memristors are described using complex differential equations. Therefore, it is necessary to integrate the various models of memristor into an unified physics-based model. In this paper, we propose a physics-informed neural network (PINN)-based compact memristor model. PINNs can solve complex differential equations intuitively and with ease. This methodology is used to conduct memristor physical analysis. The weight and bias extracted from the PINN are implemented in a Verilog-A circuit simulator to predict memristor device characteristics. The accuracy of the proposed model is verified using two memristor devices. The results show that PINNs can be used to extensively integrate memristor device models.

## 1. Introduction

The memristor was first proposed by L. Chua in the context of traditional passive devices such as resistors (R), inductors (L), and capacitors (C). This is described as the interaction between electric charge and magnetic flux [1]. The conventional von Neumann architecture faces challenges such as high-power consumption and low computing speed. Memristors can serve as integral components in neuromorphic computing to overcome these problems [2]. Recently, memristors have been the focus of significant research, particularly in their applications in various fields such as the study of non-volatile CMOS memristors [3]. Furthermore, there is active discussion on research utilizing Hopfield neural networks (HNNs) and heterogeneous discrete neural networks (HDNNs) that consider synaptic behavior [4,5].

A memristor is a resistive and non-volatile memory that changes state when a voltage is applied [6]. Essentially, a memristor device has a two-terminal metal/insulator/metal (MIM) sandwich structure [7]. In addition, memristors have various switching mechanisms depending on the material and structure such as filament formation/rupture [8], transition by amorphous/crystalline phase change [9], reversal of ferroelectric polarization direction [10], current-induced magnetization [11], etc. The physical analysis of memristors with such nonlinearity and complexity is very difficult and time-consuming [12]. The memristor I–V characteristics are described with a hysteresis curve depending on the on/off state, which, in mathematical terms, has the form of a differential equation [13]. Memristor devices have diverse and complex models depending on their structure, and there are various differential equations to solve them. In order to develop a comprehensive memristor model, researchers have tried building the framework of memristor models such as window function modification [14], hybrid memristor memory [15], capacitive connections of memristors [16], and tantalum oxide memristor models [17]. It is, therefore, not surprising that we need to unify various types of memristor models [18,19,20] for device and circuit simulators.

Solving differential equations based on traditional numerical methods, such as structural modeling of memristors and the memristor model of Messaris et al., requires significant time and effort [21,22,23,24,25,26]. To replace traditional numerical methods, the long short-term memory (LSTM) neural network methodology, a highly accurate model, is used to describe the hysteresis phenomenon of memristors. However, it has the disadvantage of having to consider past time and is more complex than the traditional multi-layer perceptron (MLP) model [27]. Moreover, the study on memristor-based neural networks utilizing recurrent neural network (RNN) [28] demonstrates excellent performance in data prediction. Nonetheless, the structure of the neural network is highly complex.

This study proposes a methodology to solve the difficult and complex differential equations of a memristor with a physics-informed neural network (PINN). PINN is an artificial neural network (ANN) for numerically solving differential equations and has a more intuitive and simpler structure than the traditional LSTM model. It can also obtain solutions to differential equations faster and more accurately than traditional models [29,30,31]. PINN can be used to predict the I–V characteristics of a memristor and extract the weight and bias of the predicted function. The neural network incorporating extracted weights and biases is implemented with Verilog-A for circuit validation. For more accurate verification, we use two memristor models. The equation for the conductance of a memristor device consists of differential equations. This implies that various differential equations exist for different structural models. PINN provides a direct and easy solution of structure-dependent physics-based differential equations. Consequently, we validate that the PINN methodology is applicable for integrating these diverse models, which has not been explored in previous memristor research. In Verilog-A, the application of the extracted weights and biases demonstrates the potential of the circuit simulator. As a result, this methodology allows for the compact modeling of various memristor devices. Additionally, it can be applied to the study of various devices that require physical analysis and integrated modeling. In this study, compact modeling has the advantage that each of the different model equations can be easily analyzed using one unified modeling technique. Accordingly, it reduces the time and cost required to construct the physical modeling of new devices.

In this work, Section 2 introduces memristor behavior and resistive state characteristics depending on voltage and time. Furthermore, we describe the model normalization procedure in PINN training. Section 3 details the PINN method, including the loss function and configuration of the neural networks. Section 4 presents the simulation process using the PINN methodology. In this section, we also illustrate the results of circuit simulator implementation in Verilog-A. Finally, Section 5 concludes the paper.

## 2. Physics Based Memristor Models

There are many different types of physics-based memristor models. The memristor behavior and resistive state characteristics depend on voltage and time. Moreover, equations for memristors are expressed in various forms. In this section, two memristor models are employed to develop a unified memristor model. The types of models are given using the generalized mean metastable switching memristor (GMMS) model [21,22] and the memristor model of Messaris et al. [23,24]. The GMMS model and the Messaris et al. model are expressed as differential equations for conductance. The GMMS model, a typical memristor model from Knowm Inc. (Santa Fe, NM, USA), has been validated for many years. Additionally, the Messaris et al. model is described as a representative memristor model that divides conductance into negative and positive values. Various memristor models, such as the HP model [13], the VTEAM model [32], the Stanford model [33], and others, are expressed in various formulations which are all described in the format of differential equations. This implies that PINN can solve any model represented as a differential equation. Therefore, we can use the two models to achieve a compact memristor model that covers a wide range of memristor devices.

### 2.1. Generalized Mean Metastable Switch (GMMS) Memristor Model

The variation of the memristor state f with time and voltage in the GMMS model can be described using a nonlinear ordinary differential equation (ODE), as represented by Equation (1). $A\left(V\right)$ and $B\left(V\right)$ are expressed by Equations (2) and (3), respectively [21,22].
(1)$$u\left(V,\;t\right)=\frac{df\left(V,t\right)}{dt}-\frac{1}{\tau }\left(A\left(V\right)\left(1-f\left(V,t\right)\right)-B\left(V\right)f\left(V,t\right)\right)$$
(2)$$A\left(V\right)=\frac{1}{1+e^{-\left(V-V_{ON}\right)/V_{t}}}$$
(3)$$B\left(V\right)=1-\frac{1}{1+e^{-(V-V_{OFF})/V_{t}}}$$
where *τ* is the memristor time constant, $V_{t}=kT/q$ ($q=1.602\times 10^{-19}$, $k=1.381\times 10^{-23},T=298.15\;K$), and V,t are voltage and time, respectively. $V_{ON}$ and $V_{OFF}$ are the switching threshold voltages of the low resistance state and high resistance state, respectively. The conductance of a memristor device with a basic structure depends on the voltage and current applied externally. The memristor conductance $G\left(f,V\right)$ and current I by Equation (1) are expressed as follows [21,22]
(4)$$I=G\left(f,V\right)V,\;\;\;\;\;\;G(f,V)=\frac{f(V,t)}{R_{ON}}+\frac{1-f(V,t)}{R_{OFF}}$$
where $R_{ON}$ and $R_{OFF}$ are the resistors in the on/off state.

### 2.2. Memristor Model of Messaris et al.

The variation of the resistance (R) with time and voltage in the memristor model of Messaris et al. is expressed as follows [23,24]:(5)$$\frac{dR(V,\;t)}{dt}=s_{p,n}\left(V\right)g_{p,n}\left(V,t\right)$$
where $s_{p,n}\left(V\right)$ is the switching sensitivity which depends on the applied voltage V and is represented by
(6)$$s_{p,n}\left(V\right)=\left\{\begin{array}{c} a_{p}(-1+e^{\left|V\right|/t_{p}}),\;\;\;\;\;\;for\;V>0\;\\ a_{n}(-1+e^{\left|V\right|/t_{n}}),\;\;\;\;\;\;for\;V\leq 0 \end{array}\right.$$
with the fitting parameters $a_{p}$, $a_{n}$, $t_{p}$, and $t_{n}$. p and n denote positive and negative voltage values, respectively. $g_{p,n}\left(V,t\right)$ is the window function which depends on time and voltage, and is given by [25,26]
(7)$$g_{p,n}\left(V,t\right)=\left\{\begin{array}{c} {\left(r_{p}-R\right)}^{2}\;\;\;\;\;\;\;\;\;\;\;\;\;\;\;\;\;\;\;\;for\;V>0\\ {\left(R-r_{n}\right)}^{2}\;\;\;\;\;\;\;\;\;\;\;\;\;\;\;\;\;\;\;for\;V\leq 0 \end{array}\right.$$
where $r_{p,\;n}(V)$ is the absolute threshold function represented by [23,25]
(8)$$r_{p,\;n}\left(V\right)=\left\{\begin{array}{c} r_{p}=r_{p0}+r_{p1}V\;\;\;\;\;\;\;\;\;\;\;\;\;for\;V>0\\ r_{n}=r_{n0}+r_{n1}V\;\;\;\;\;\;\;\;\;\;\;\;\;for\;V\leq 0 \end{array}\right.$$
with the fitting parameters $r_{p0}$, $r_{p1}$, $r_{n0}$, and $r_{n1}$. In particular, the parameters $r_{p1}$ and $r_{n1}$ determine the rate at which they change in response to the applied voltage. The current equation as a function of time and voltage can be given by
(9)$$I\left(R,V\right)=\left\{\begin{array}{c} A_{p}(1/R)\sinh({\;B}_{p}V)\;\;\;\;\;\;\;\;for\;V>0\;\\ A_{n}\left(1/R\right)\sinh\left({\;B}_{n}V\right)\;\;\;\;\;\;\;for\;V\leq 0 \end{array}\right.$$
where $A_{p}$, $A_{n}$, $B_{p}$, and $B_{n}$ are fitting parameters [23].

For the memristor model of Messaris et al., the resistance range is $R_{min}\leq R\leq R_{max}$, where $R_{min}$ and $R_{max}$ represent the minimum and maximum values of resistance (R). The normalization of the memristor model of Messaris et al. turns resistance to resistive state and is obtained by performing the min-max feature scaling method. The min-max feature scaling is used to convert values in the R range to [0,1] data. Furthermore, the normalization is helpful to prevent overfitting in neural network training. The memristor state f from the min-max feature scaling is represented as follows [34,35]:(10)$$f\left(V,t\right)=\frac{R-R_{min}}{{R_{max}-R}_{min}}=\frac{R-R_{min}}{R_{dev}}$$
where $R_{dev}$ is the deviation between the maximum and minimum values of R. Using Equations (5) and (10), the link between the memristor state and the resistance is expressed as follows:(11)$$u\left(V,t\right)=\frac{df\left(V,\;t\right)}{dt}-\left(\frac{1}{R_{dev}}\right)s_{p,n}\left(V\right){\left(r_{p,\;n}\left(V\right)-R_{min}-R_{dev}f\left(V,\;t\right)\right)}^{2}$$

This normalization procedure not only significantly speeds up the computation but also reduces the error [34,35].

## 3. Physics-Informed Neural Network Model

The PINN is an artificial neural network designed to numerically solve physical laws described using general nonlinear partial differential equations [31]. PINN is developed to predict the behavior of systems according to physical laws even when data are scarce or incomplete. The neural network learns to enforce physical laws for given initial and boundary conditions, enabling accurate and efficient predictions. Moreover, it learns faster and operates effectively in complex systems compared to traditional numerical analysis methods. The memristor models of I–V characteristics produce hysteresis, which is represented using a complex differential equation [13]. Therefore, the physical modeling of the memristor is analyzed using PINN. In this section, we use PINN to learn the state differential equations corresponding to Equations (1) and (11) to analyze the dynamics of the memristor.

Given a function $N\left(w,\;b\right)$ that has undergone PINN training, the final solution with the initial condition C can be represented as follows [29]:(12)$$f\left(V,t\right)=C+N(w,\;b)$$
where w and b are the weight and the bias, respectively. The loss optimization function for a typical ANN is represented as follows:(13)$$L_{ANN}=\left\|f\left(V,t\right)-N(w,\;b)\right\|$$

However, PINN training is conducted using unsupervised learning where no solution exists. The loss function (L) of PINN unsupervised learning can be expressed by
(14)$$L=\left\{\begin{array}{c} L_{ODE,\;A}+L_{IC}\;\;\;\;\;\;\;\;\;\;\;\;for\;GMMS\;model\\ L_{ODE,\;B}+L_{IC}\;\;\;\;\;\;\;\;\;\;\;\;for\;Messaris\;model \end{array}\right.$$
where $L_{ODE}=(1/N_{ODE})\sum_{i=1}^{N_{ODE}}{\left(u(V_{i},t_{i})\right)}^{2}$ signifies the mean square error (MSE) on the residuals of the ODE [29,31]. $N_{ODE}$ is the number of input data and $L_{IC}$ is denoted as MSE between the initial conditions for state training value at its coordinates. In the case of the GMMS model, L is computed using $L_{ODE,\;A}$. For the Messaris et al. model, it is calculated using $L_{ODE,\;B}$. $L_{ODE,\;A}$ represents the $L_{ODE}$ for the GMMS model calculated by using Equation (1), and $L_{ODE,\;B}$ corresponds to the $L_{ODE}$ for the Messaris model obtained by applying Equation (11). $L_{ODE,\;A}$ and $L_{ODE,\;B}$ are given by
(15)$$L_{ODE,A\;\;}=\frac{1}{N_{ODE}}\sum_{i=1}^{N_{ODE}}{\left(\frac{df\left(V_{i},t_{i}\right)}{dt}-\frac{1}{\tau }\left(A\left(V_{i}\right)\left(1-f\left(V_{i},t_{i}\right)\right)-B(V_{i})f\left(V_{i},t_{i}\right)\right)\right)}^{2}$$
(16)$$L_{ODE,B}=\frac{1}{N_{ODE}}\sum_{i=1}^{N_{ODE}}{\left(\frac{df\left(V_{i},t_{i}\right)}{dt}-\left(\frac{1}{R_{dev}}\right)s_{p,n}\left(V_{i}\right){\left(r_{p,\;n}\left(V_{i}\right)-R_{min}-R_{dev}f\left(V_{i},t_{i}\right)\right)}^{2}\right)}^{2}$$
and $L_{IC}$ for state f and training value $\hat{f_{i}}$ is represented by [29,30,31]
(17)$$L_{IC}=\frac{1}{N_{IC}}\sum_{i=1}^{N_{IC}}{\left(f\left(V_{i},\;t_{i}\right)-\hat{f_{i}}\right)}^{2}$$
where $N_{IC}$ is the number of the initial condition [29,30,31].

The PINN structure and Verilog-A implementation are shown in Figure 1, in which the initial condition C=0 for input data V= −0.2~0.2 and t = 0~0.1 and the ODE of Equation (1) are trained. Here, the hidden layer consists of two layers, each with 10 neurons. It uses the MLP neural network and is trained with a sigmoid function. Similarly, we can apply the same methodology to the memristor model of Messaris et al. It is trained with an initial condition C=0 and the ODE of Equation (11) for input data V = −1~1 and t = 0~0.1. Furthermore, the hidden layer is composed of two layers, each with 20 neurons. The sigmoid function is used as the activation function for the hidden layer.

The weight and bias operations for each layer are represented by
(18)$$h_{j}^{\left(1\right)}=\sum_{i}\frac{1}{1+e^{-(w_{1i}^{\left(1\right)}*V+w_{2i}^{\left(1\right)}*t+b_{i}^{\left(1\right)})}}$$
(19)$$h_{j}^{\left(2\right)}=\sum_{i}\left(\sum_{j}\frac{1}{1+e^{-\left(w_{ji}^{\left(2\right)}*h_{j}^{\left(1\right)}+b_{i}^{\left(2\right)}\right)}}\right)$$
(20)$$N\left(w,b\right)=\sum_{j}w_{j}^{\left(3\right)}h_{j}^{\left(2\right)}+b_{out}^{\left(3\right)}$$
where $h_{j}^{\left(k\right)}$ is the output of the jth neuron in the kth hidden layer. In MLP, the output of one hidden layer is passed to the next hidden layer input. First, the first hidden layer parameter configuration is as follows: $w_{1i}^{\left(1\right)}$ is the ith weight of input 1, $w_{2i}^{\left(1\right)}$ is the ith weight of input 2, and $b_{i}^{\left(1\right)}$ is the ith bias value. In the second hidden layer parameter configuration, $w_{ji}^{\left(2\right)}$ is the ith weight of the jth neuron, and $b_{i}^{\left(2\right)}$ is the ith bias value. Finally, the output layer parameter configuration is as follows: $w_{j}^{\left(3\right)}$ is the weight of the output layer jth neuron, and $b_{out}^{\left(3\right)}$ is the output layer bias value [36,37].

## 4. Results and Discussion

A PINN compact model is implemented using the two models described in Section 2. The training model of PINN is constructed using the python package SciANN [38]. The neural network is composed of two layers with 10 neurons and 20 neurons for each of the two models. The use of a simple neural network structure implies a significant reduction in computer execution time. Furthermore, the Verilog-A simulation takes a relatively short time as it operates simply with the weights and biases extracted from the neural network. The simulation process unfolds as follows: 1. Train the two models outlined in Section 2 through Python-based PINN learning. 2. Extract weight and bias values from the trained results. 3. Generate symbols in Verilog-A using the extracted weight and bias values. 4. Implement the circuit using the generated symbols. 5. Verify whether the physical characteristics obtained through traditional numerical analysis methods and PINN learning are identical. For GMMS model, the parameters required for training the GMMS model are provided in Table 1 [21].

The most difficult part of training the memristor state through PINN is insufficient training in regions where the time axis is close to 0 and the voltage axis is at inflection points. To increase the training accuracy on the time axis, we further divided the time input from 2000 intervals to 5000 intervals from 0 to 0.1. We also reduced the training rate from 0.1 to 0.001, which helped improve accuracy. Since this is unsupervised learning, the presented initial conditions must also be correct. When Equation (1) is trained by PINN, this can produce a memristor state as demonstrated in Figure 2. Figure 2a is the state described by the numerical solution data of Equation (1), and Figure 2b is the state produced from PINN training data. In the GMMS model, the memristor state approaches 1 as time and voltage increase.

Figure 3a,b show the error between Figure 2a,b, which are numerical solution data and PINN training data, respectively. The calculated maximum error is 0.018, and the MSE is $5\times 10^{-6}$, giving a training accuracy of 98.5%.

The loss function is a measure of how well the neural network is performing during training. The loss function is calculated via Equations (14), (15) and (17). The loss function in Figure 4 shows a significant reduction in loss in the early epochs and is optimized by the optimization algorithm. Finally, a loss of $6\times 10^{-7}$ is obtained, indicating that learning is being performed very well.

The weights and biases are extracted through the PINN learning method. These values are used to predict the I–V characteristic curve by substituting them into Equations (18)–(20). To calculate the current and voltage, the conductance of the memristor was calculated first, as shown in Figure 5. The input of simulation is a sin waveform with an amplitude of 0.2 for 0.1 s at different frequencies 10 Hz, 100 Hz, and 1 kHz.

Figure 6 shows the I–V characteristic curves obtained from Verilog-A circuit simulation [23,39]. The dotted line represents the numerical solution data and the solid line represents the PINN training data. Figure 6a–c show the results when the input signals have a frequency of 10 Hz, 100 Hz and 1 kHz, respectively. Accurate I–V characteristic curves were predicted for input frequencies under various conditions. The current–voltage characteristics of the GMMS model can be influenced by variations in $R_{ON}$ and $R_{OFF}$, as shown in Table 1. If $R_{ON}$ and $R_{OFF}$ decrease, the set current and reset current increase, respectively.

The simulation for the memristor model of Messaris et al. is performed. Table 2 presents the necessary parameters for training the memristor model of Messaris et al. [23].

The memristor model of Messaris et al. uses PINN to train the positive and negative regions, respectively. When the differential equations for resistance in the memristor model of Messaris et al. are solved using PINN, there is an issue with values becoming excessively large. This disrupts the proper progress of training in the neural networks. To address this issue, we proceed with training by applying the normalization method mentioned in Section 2. This method transforms resistance into an equation related to the resistive state. The resistive state in the positive region is illustrated in Figure 7. Figure 7a,b represent numerical solution data and PINN training data, respectively.

In the positive region, the error between the numerical solution data in Figure 7a and the PINN training data in Figure 7b is depicted in Figure 8. There are many inflection points, but the maximum error is very small at 0.001.

The resistive state in the negative region is depicted in Figure 9. Figure 9a represents the numerical solution data and Figure 9b represents the PINN training data.

Figure 10 shows the error between the numerical solution data and PINN training data in the negative region. The maximum error rate is 0.006. In contrast to the positive region, the negative region exhibits more inflection points.

The PINN training loss for the memristor model of Messaris et al. is depicted in Figure 11. Figure 11a,b represent the learning loss in the positive region and the negative region, respectively. The memristor model of Messaris et al. was set up with 1000 epochs and a learning rate of 0.01. The loss for the positive region and the negative region converges to the level of approximately $10^{-6}$ and $10^{-4}$, respectively.

The simulation uses the same input as the GMMS model, consisting of a 10 Hz, 100 Hz, and 1 kHz sin waveform. The conductance can be calculated using the current equation of Equation (9) for the memristor model of Messaris et al., as illustrated in Figure 12. Other models [19] show the same magnitude of conductance in the positive and negative regions for a sin waveform input. However, the magnitude of the positive and negative regions is not equal due to the four fitting parameters given in Equation (9) of the memristor model of Messaris et al.

The I–V characteristic curve for the memristor model of Messaris et al. is depicted in Figure 13. The difference between the cases at 10 Hz and 100 Hz frequencies is not observed clearly. However, it is possible to analyze the differences by enlarging the I–V characteristic curve, as shown in the inset pictures of Figure 13. At 10 Hz, one hysteresis curve is generated, while the number of hysteresis curves at 100 Hz and 1 kHz is ten and one hundred, respectively. To verify accurate current predictions at high frequencies, the square pulses with a frequency of 1 MHz (pulse width 1 μs) are applied to the GMMS and Messaris models, as shown in Figure 14a,b. It can be observed that accurate current predictions are achieved even with high frequency square pulses.

Stochastic non-ideal properties of memristors, such as noise [40] and variability [41], can be modeled in the form of differential equations [42,43]. The methodology in this paper can facilitate the stochastic modeling of memristors.

## 5. Conclusions

To date, there has been much research on memristors, but no generic model has been proposed. In this study, we propose a methodology for training memristor state ODEs using the unsupervised learning of PINNs. This methodology has a simpler structure and better prediction performance than conventional neural networks. This can solve traditional memristor state ODEs that are complex to analyze numerically, which allows the integration of many existing memristor device models and enables a compact model for circuit simulator. Two models are presented to validate the memristor compact modeling methodology. The resistive states of the two models are analyzed using PINN learning. The trained data can be used to predict the physical properties of memristors with high accuracy. This proves that the memristor compact modeling methodology using PINN is reasonable. In addition to presenting a PINN-based compact model under ideal conditions, the inclusion of variability in the PINN memristor model is essential. Future research will include a PINN methodology for modeling non-ideal device characteristics such as device-to-device variability and cycle-to-cycle variability.

## Figures and Tables

**Figure 1 micromachines-15-00253-f001:**
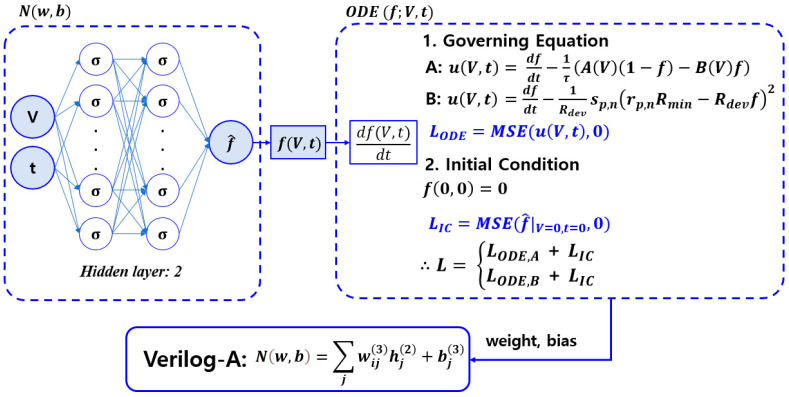
PINN and Verilog-A implementation methodology. The hidden layer computation is performed via weight and bias for the input data *V* and *t*. The weight and bias obtained by PINN are implemented into a Verilog-A script.

**Figure 2 micromachines-15-00253-f002:**
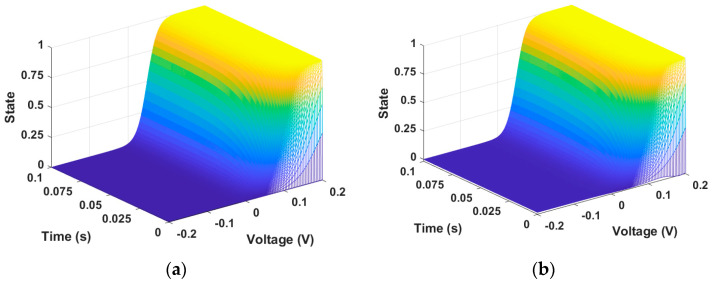
Memristor state. (**a**) Numerical solution data and (**b**) PINN training data. The state changes according to the input data V= −0.2~0.2 and t = 0~0.1. The state change is represented by a value between 0 and 1.

**Figure 3 micromachines-15-00253-f003:**
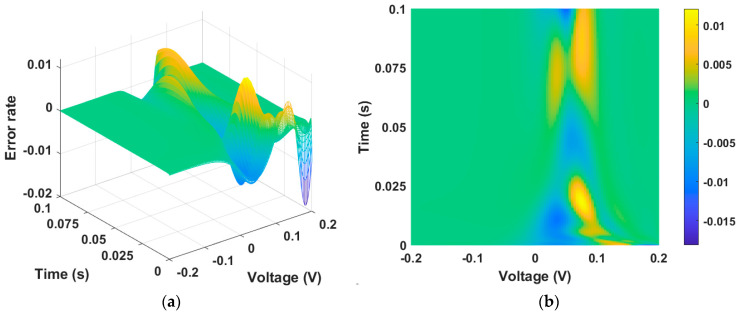
Error between numerical solution data and PINN training data. (**a**) Three-dimensional plot and (**b**) two-dimensional plot. The error is largest at the point where the time axis is zero, as well as at the inflection point of the state function.

**Figure 4 micromachines-15-00253-f004:**
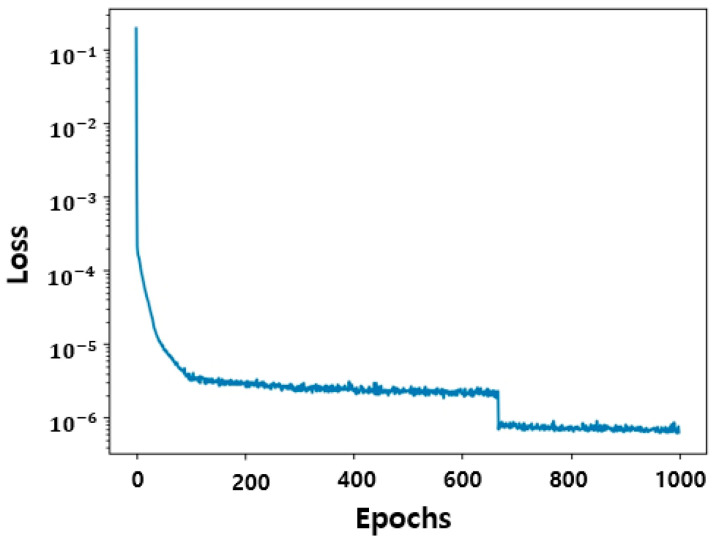
History of PINN loss values. The learning rate and the number of iterations are fixed to 0.01 and to 1000, respectively. The loss function is computed up to 1000 epochs and converges at $6\times 10^{-7}$.

**Figure 5 micromachines-15-00253-f005:**
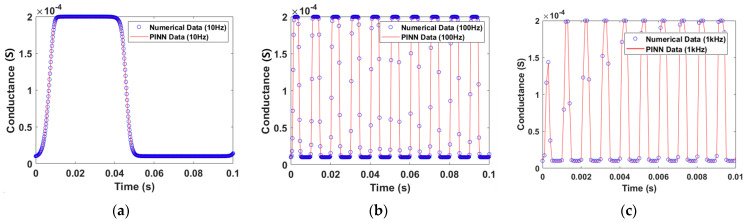
The GMMS Model conductance. (**a**) Input signal of a 10 Hz sin wave with an amplitude of 0.2 V. (**b**) Input signal of a 100 Hz sin wave with an amplitude of 0.2 V. (**c**) Input signal of a 1 kHz sin wave with an amplitude of 0.2 V.

**Figure 6 micromachines-15-00253-f006:**
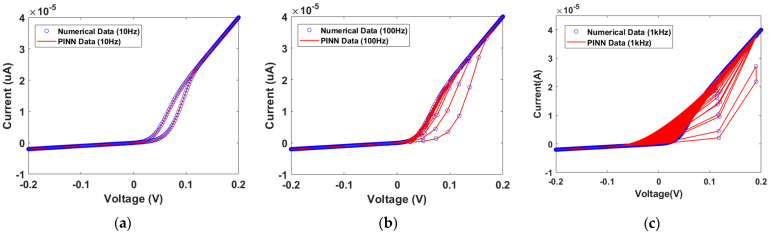
Numerical solution data and PINN-predicted I–V characteristic curves from the GMMS model. (**a**) Input frequency at 10 Hz sin wave with an amplitude of 0.2 V. (**b**) Input frequency at 100 Hz sin wave with an amplitude of 0.2 V. (**c**) Input frequency at 1 kHz sin wave with an amplitude of 0.2 V.

**Figure 7 micromachines-15-00253-f007:**
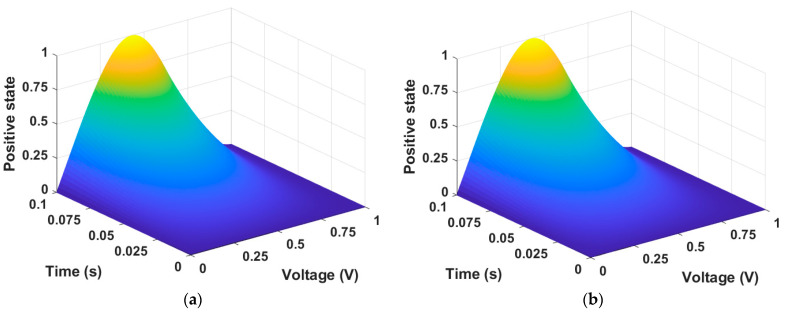
Resistive state of the positive region in the memristor model of Messaris et al. (**a**) Numerical solution data and (**b**) PINN training data. The state changes according to the input data V = 0~1 and t = 0~0.1. The state change is represented by a value between 0 and 1.

**Figure 8 micromachines-15-00253-f008:**
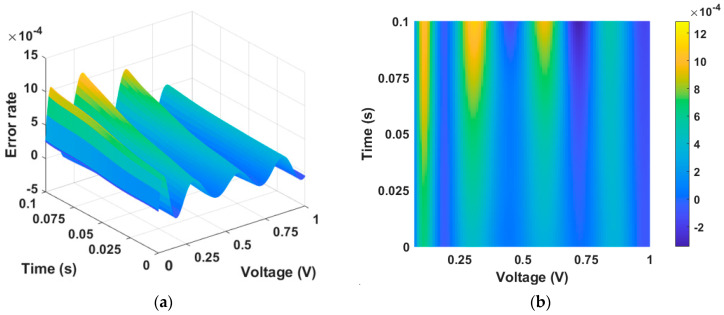
Error rate between numerical solution data and PINN training data in the positive region in the memristor model of Messaris et al. (**a**) Three-dimensional plot and (**b**) two-dimensional plot. The error is largest at the point where the time axis is zero, as well as at the inflection point of the state function.

**Figure 9 micromachines-15-00253-f009:**
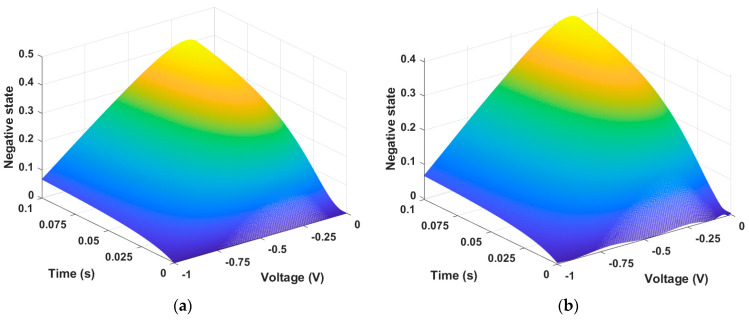
Resistive state of the negative region in the memristor model of Messaris et al. (**a**) Numerical solution data and (**b**) PINN training data. The state changes according to the input data V = −1~0 and t = 0~0.1. The state change is represented by a value between 0 and 1.

**Figure 10 micromachines-15-00253-f010:**
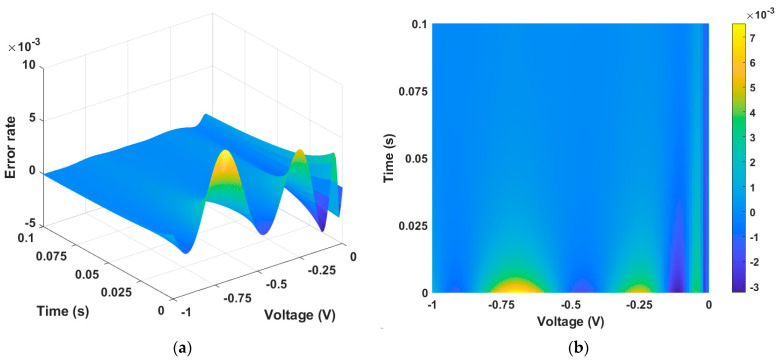
Error rate between numerical solution data and PINN training data in the negative region in the memristor model of Messaris et al. (**a**) Three-dimensional plot and (**b**) two-dimensional plot. The error is largest at the point where the time axis is zero, as well as at the inflection point of the state function.

**Figure 11 micromachines-15-00253-f011:**
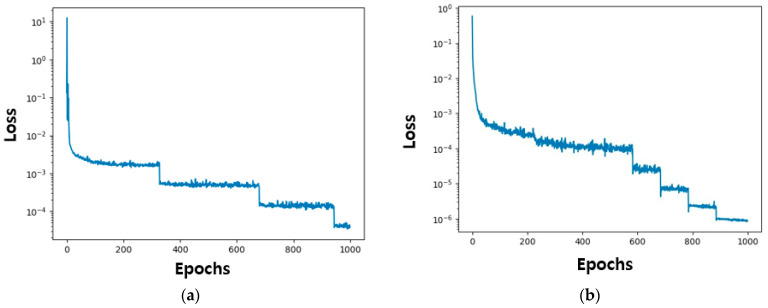
History of PINN loss function results: (**a**) 1000 epochs in the positive region and (**b**) 1000 epochs in the negative region.

**Figure 12 micromachines-15-00253-f012:**
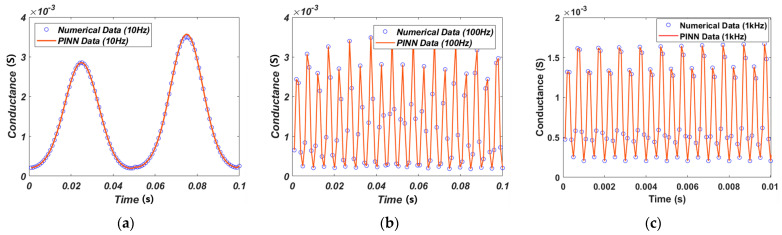
The memristor model of Messaris et al. conductance. (**a**) Input signal of a 10 Hz sin wave with an amplitude of 1 V. (**b**) Input signal of a 100 Hz sin wave with an amplitude of 1 V. (**c**) Input signal of a 1 kHz sin wave with an amplitude of 1 V.

**Figure 13 micromachines-15-00253-f013:**
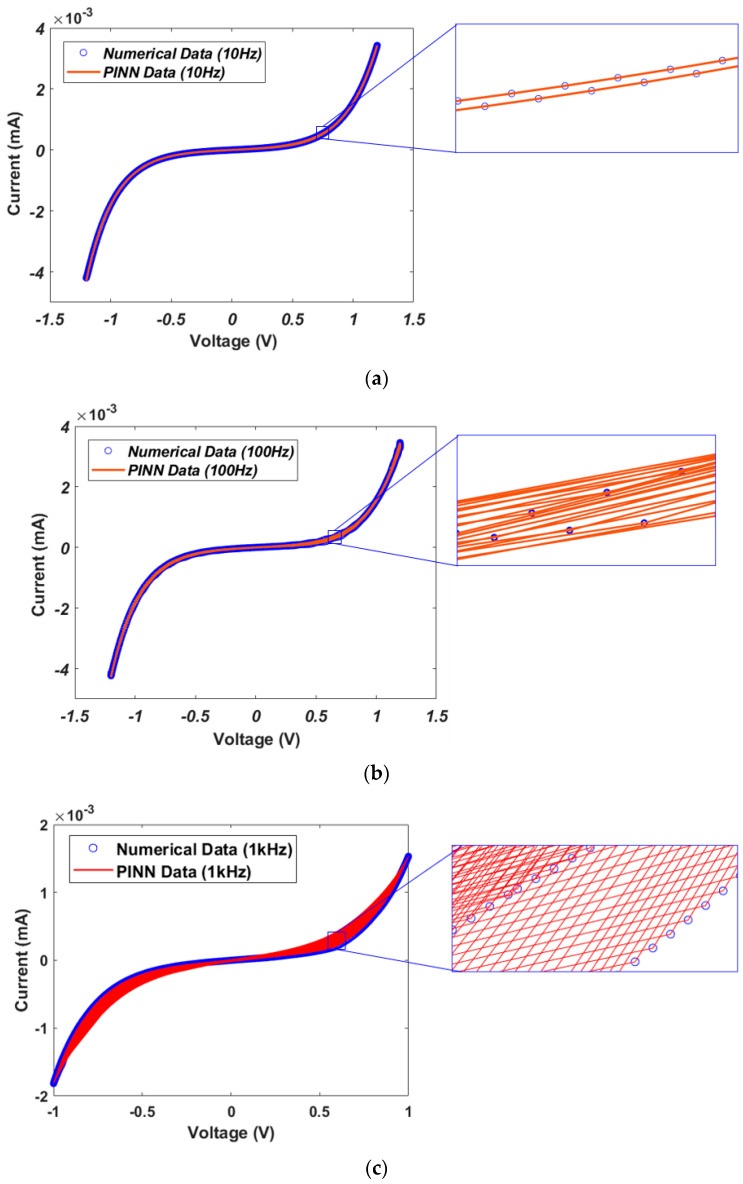
I–V characteristic curves between numerical data and PINN data. (**a**) Input signal of a 10 Hz sin wave with an amplitude of 1 V. (**b**) Input signal of a 100 Hz sin wave with an amplitude of 1 V. (**c**) Input signal of a 1 kHz sin wave with an amplitude of 1 V.

**Figure 14 micromachines-15-00253-f014:**
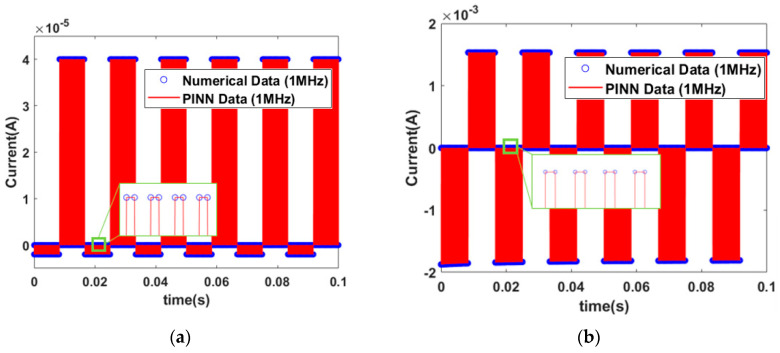
The memristor model of I–V characteristic curve. (**a**) Input signal of a pulse wave with a frequency of 1 MHz in a GMMS model. (**b**) Input signal of a pulse wave with a frequency of 1 MHz in the memristor model of Messaris et al.

**Table 1 micromachines-15-00253-t001:** Parameters for PINN training of GMMS models.

V	$R_{ON}(\Omega )$	$R_{OFF}(\Omega )$	$V_{ON}(V)$	$V_{OFF}(V)$	τ
−0.2≤V<0.2	5000	100,000	0.2	0.1	0.0001

**Table 2 micromachines-15-00253-t002:** Parameters for PINN training in the memristor model of Messaris et al.

V	$a_{p,n}$	$t_{p,n}$	$r_{p0,n0}$	$r_{p1,n1}$	$R_{min}(\Omega )$	$R_{max}(\Omega )$
V>0	0.01	2.45	71.61	4370	4513	7000
V≤0	−0.52	2.72	6006	1279	4513	7000

## Data Availability

The data presented in this study are available on request from the corresponding author.

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
