# Peer review of "A Compact Memristor Model Based on Physics-Informed Neural Networks"

_micromachines, 2024, doi:10.3390/mi15020253_

Round 1

Reviewer 1 Report

Comments and Suggestions for Authors

Review for micromachines 2821865

The paper is interesting and worth’s publication. However, the authors should take into account the following points

1)        Some references should be included

a.     Sha, Y., Ouyang, L., & Chen, Q. (2021, July). A Physics-Informed Neural Network for RRAM modeling. In 2021 International Applied Computational Electromagnetics Society (ACES-China) Symposium (pp. 1-2). IEEE.

b.     Tang, Z., Sun, B., Zhou, G., Zhou, Y., Cao, Z., Duan, X., ... & Shao, J. (2023). Research progress of artificial neural systems based on memristors. Materials Today Nano, 100439.

2)        Could the authors should discuss on the code/software they employed for the simulations?

3)        Some minor grammatical errors e.g.

a.        line 185 “states shown” should be “states are shown”.

b.        Line 200 “values extracted” should be” “values are extracted”

c.        Please check for any other similar errors

4)        Fig. 6 the authors should explain better the behavior observed in the figures (are they function of a given parameter?

5)        Could the authors provide details about the computer execution times?

Comments on the Quality of English Language

There are some minor correction. Some are mentioned in the review comments

Reviewer 2 Report

Comments and Suggestions for Authors

This manuscript aims to integrate various memristor models into a unified physics-based model. The authors proposed a PINN-based compact model to integrate memristor device models. However, the following considerations should be addressed.

1)     What is the significance of this proposed PINN method? The mathematical mode of a memristor is obtained by function fitting after measuring the electrical characteristics of the real device. The authors propose to use neural network to lean the existing memristor mathematical models. What are the benefits of this work on memristor modeling?

2)     What is the significance of studying a unified memristor model? Whether it is good for memristor-based circuit design of for memristor-based device manufacturing? Since each memristor is implemented based on different physical mechanisms, why propose a unified memristor model?

3)     The authors aim to establish a unified mathematical model of memristor, but only to models are used for method verification. This simple experiment may lead to the unreliability of the results. It is suggested that the authors add more memristor model to ensure the reliability of the neural network results.

4)     In (14) the loss function L has two expressions, where Lodea+Lic and Lodeb+Lic. What is the condition for L to choose which expression?

5)     Real memristor devices contain many non-ideal characteristics, and it is difficult to model memristors with these non-ideal characteristics. Why do not include the non-ideal characteristics, such as noise, drift, randomness into this research?

6)     Some latest references should be citedsuch as 1Dynamics analysis, synchronization and FPGA implementation of multiscroll Hopfield neural networks with non-polynomial memristor, Chaos, Solitons & Fractals, vol. 179, Article ID 114440, 2024;【2Dynamical Behavior of MemristorCoupled Heterogeneous Discrete Neural Networks with Synaptic Crosstalk. Chinese Phys. B2023;【3Nonvolatile CMOS memristor, reconfigurable array, and its application in power load forecasting, IEEE Transactions on Industrial Informatics, 2023, Early Access, DOI: 10.1109/TII.2023.3341256

Round 2

Reviewer 2 Report

Comments and Suggestions for Authors

The revised paper has addressed the questions, and can be acceptable.